# Effects of Ozone Application on Microbiological Stability and Content of Sugars and Bioactive Compounds in the Fruit of the Saskatoon Berry (*Amelanchier alnifolia* Nutt.)

**DOI:** 10.3390/molecules27196446

**Published:** 2022-09-30

**Authors:** Józef Gorzelany, Ireneusz Kapusta, Miłosz Zardzewiały, Justyna Belcar

**Affiliations:** 1Department of Food and Agriculture Production Engineering, University of Rzeszow, 4 Zelwerowicza Street, 35-601 Rzeszów, Poland; 2Department of Food Technology and Human Nutrition, University of Rzeszow, 4 Zelwerowicza Street, 35-601 Rzeszów, Poland

**Keywords:** cultivars and clones of Saskatoon berry fruits, ozonation process, microbiological stability, acidity of Saskatoon berry fruits, sugars, bioactive compounds, polyphenols profile, ascorbic acid, antioxidant activity

## Abstract

Saskatoon berry fruits are a valuable source of micro- and macronutrients, sugars, and compounds with health-promoting properties, the properties of which change during storage. This study presents the effects of applied gaseous ozone at 10 ppm for 15 and 30 min on microbiological stability, sugar content, and bioactive compounds for three cultivars and three clones of Saskatoon berry fruit. The ozonation process had a positive effect on reducing the microbial load of the fruit, which was observed on day 7 of storage for the two variants of ozonation time of 15 and 30 min compared to the control and also on the sugar profile of the “Thiessen” fruit, as well as clones no 5/6 and type H compared to the control sample, which was non-ozonated fruit. In the Saskatoon berry fruits analyzed, 21 polyphenolic compounds were identified, of which four belonged to the anthocyanin group whose main representative was the 3-O-glucoside cyanidin. The ascorbic acid content and antioxidant activity (determined by DPPH· and ABTS^+^· methods) varied according to the cultivar and clone of the Saskatoon berry fruits analyzed and the ozone exposure time.

## 1. Introduction

The growing awareness of health-promoting consumers has led the food industry to seek new raw materials with nutritional, antioxidant, and health-promoting properties. In recent years, the Saskatoon berry (*Amelanchier alnifolia* Nutt.) has become of increasing scientific interest. These are blue-purple, oval-shaped fruits characterized by a high content of sugars, organic acids, minerals, and polyphenols. Due to their chemical composition, fruits can be used in the production of jams, jelly, wines, or syrups [1,2,3]. Saskatoon berry fruits can be used in the production of functional beverages due to the stability of the syrup and its antioxidant, flavor, or coloring properties [4]. Furthermore, the fruit can be used as a natural food coloring due to its high anthocyanin content [5].

The high content of sugars (mainly glucose, fructose, sucrose, and sorbitol) and phenolic compounds contained in the fruits of the Saskatoon berry are responsible for the sensory properties of the raw materials and their attractiveness desired by consumers [6]. The fruit peel contains anthocyanins (cyanidin derivatives), flavonoid compounds, including rutin, hyperoside, avixene, or quercetin [2,7], phenolic acids including 3-feruloylquinic, chlorogenic, and 5-feruloylquinic acids [8]. Polyphenols contained in the Saskatoon berry exhibit strong antioxidant, anti-inflammatory, or anti-allergic effects [2,9]. The anticancer effects of the fruit are by counteracting changes that occur during oxidative stress and inflammation, and by regulating carcinogenic and xenobiotic metabolizing enzymes, various transcription and growth factors, inflammatory cytokines, and subcellular signaling pathways of cancer cell proliferation, apoptosis, and tumor angiogenesis [10]. The fruits of Saskatoon berry also have antidiabetic activity. A powder obtained from the berries containing cyanidin-3-galactoside and cyanidin-3-glucoside reduces glucose, insulin, and blood lipid levels [11].

Ozonation technology that extends the shelf life of food is considered a nonthermal method of food preservation that improves food safety without compromising food quality or polluting the environment [12]. The disinfectant and biocidal properties of ozone have attracted the interest of the fruit and vegetable sector because the ozone molecule rapidly decomposes into oxygen, leaving no residue on the product. Ozone reacts with some organic food compounds, resulting in the formation of possible byproducts, e.g., aldehydes, ketones, or carboxylic acids, which pose no risk to human health [13]. Several studies have shown that aqueous ozone is a good alternative to traditional sterilizers, as it destroys harmful bacteria and viruses even at low concentrations [14]. Ozone inactivates up to 99.0% of pesticides and most microorganisms present in various food tissues, due to its potential oxidizing capacity [15]. Ozone has a positive effect on ascorbic acid levels, which can be explained by the stress generated by the oxidative molecule and the increased synthesis and accumulation of antioxidant compounds. Ummat et al. [16] showed that ozonated water reduced the microbial load on pepper fruits, which also had a higher ascorbic acid content as a result of this treatment compared to the control sample. Similar trends were observed for freshly harvested pineapple fruit treated with aqueous ozone at 0.6, 0.9 and 1.5 ppm and stored for 20 days at 2 °C [17]. Papaya fruit after 20 min of ozone treatment showed an increase in phenolic content in the fruit by 10.3%, while ascorbic acid content decreased by 2.3% compared to untreated fruit. Furthermore, gaseous ozone reduced the number of microorganisms, i.e., coliforms and mesophilic bacteria [18].

The purpose of this study was to determine the microbial stability of the ozone treated fruit of three cultivars and three prospective clones of the Saskatoon berry and the effect of ozone treatment on the content of sugars and bioactive compounds in the Saskatoon berry fruit.

## 2. Results and Discussion

### 2.1. Changes in the Microbiological Properties of the Ozone Treated Saskatoon Berry Fruit

The storage life of fruit is significantly dependent on the content of microorganisms on the fruit surface, which activate unfavorable biochemical transformations in the fruit, causing a change in selected chemical properties, thereby deteriorating the quality of the raw material. Ozone is an abiotic factor that causes damage to the metabolism of microorganisms in the fruit, thus causing an increase in its storage life [19].

A high load of mesophilic aerobic bacteria was observed on the first day after ozonation of the fruit of the Saskatoon berry. For the analyzed cultivars, a positive effect of the process on the reduction in the abundance of the bacteria was observed for the analyzed fruit cultivars on the first day after ozonation, irrespective of the dose applied. During the 10th day of storage, the highest mesophilic aerobic bacteria were recorded for the control sample (Table 1). In the case of the fruit ozone variant applied, the dose of ozone at 30 min had the most beneficial effect on reducing microbial counts. For the varieties analyzed, the lowest values of microbial load were observed on the 7th day of storage for both variants ozonated for 15 and 30 min compared to the control (Table 1). An increase in the abundance of aerobic mesophilic bacteria was recorded on the 10th day of storage for each variant of the experiment compared to 7th days. The lowest values of microbial load were observed for the ‘Honeywood’ cultivar and the clone type N on 7th days of storage for the variant of ozonated fruit for 30 min (Table 1). The application of gaseous ozone slowed the increase in the abundance of the bacteria studied, whose presence in the fruit contributes to storage losses. Similar relationships were observed for blueberries stored in a gaseous ozone atmosphere. The ozonation process for these fruits almost completely stopped the development of grey mold [20]. Ayranci et al. [21] found an effect of ozonation on the quality parameters of turkey meat. After the application of gaseous ozone to the raw material studied, a reduction in the number of all aerobic mesophilic bacteria and a high efficiency in the deactivation of microorganisms were observed. Pretell-Vásquez et al. [22] applied gaseous ozone at a concentration of 6–10 ppm for 20 min to harvested asparagus. Based on their study, they observed its effect on reducing microbial counts, in particular the number of mesophilic aerobic bacteria and molds over a 25-day storage period for this raw material. Zardzewiały et al. [23] treated rhubarb petioles with gaseous ozone concentrations ranging from 10 to 100 ppm for 5–30 min. The application of the proposed combinations of doses and ozonation times resulted in a significant reduction in the cfu of bacteria tested by at least 1 log cfu·g^−1^.

During storage, a high burden of yeast and molds was observed in the fruits of Saskatoon berry. The positive effect of the ozonation process, regardless of the dose applied, on reducing the microbial load on the fruit was observed in the analyzed terms of the tests performed. During the storage period, the study found that the highest incidence of yeast and molds occurred on the fruit of the control trial. In the case of ozonized varieties, both the application of ozone at 15 and 30 min reduced the tested microbial load compared to the control. For these variants, the lowest yeast and mold infestation were observed on the 7th days of storage (Table 2). On the 10th day of storage, the lowest microbial infestation was characterized by the ‘Honeywood’ variety that ozonated for 30 min compared to the other varieties. Based on the tests carried out on the 10th day of storage, it was found that ozonation with a dose of 10 ppm for 30 min reduced the number of yeasts and molds for the varieties tested by an average of 0.55 log cfu·g^−1^ compared to the control (Table 2). In contrast, for the same ozone dose applied for 15 min, there was a reduction in the number of microorganisms tested by an average of 0.48 log cfu·g^−1^ compared to the control variant (Table 2.). Aguayo et al. [24] report that exposure to cyclic ozone (every three hours) reduces the number of aerobic mesophilic aerobic bacteria by 1.1 log cfu·g^−1^ and fungi by 1.75 log cfu·g^−1^ in tomatoes compared to control samples not treated with ozone. Matłok et al. [25] fumigated marjoram plants with gaseous ozone at a dose of 1 ppm. This treatment resulted in a significant decrease in the number of yeast and molds on the first and fifth day after fumigation. The best result was observed when the marjoram plants were exposed to ozone for 10 min. According to Horvitz et al. [26] the ozonation of freshly harvested red peppers reduced the number of mesophilic bacteria by up to 2.6 log cfu·g^−1^ compared to the control. Zapalowska et al. [19] treated sea buckthorn berries with gaseous ozone at a concentration of 100 ppm for 30 min. The process conditions used resulted in a reduction in aerobic bacterial colonies by 3 log cfu·g^−1^ compared to the control (non-ozonated sample).

### 2.2. Changes in pH and Acidity in the Saskatoon Berry during Ozone Exposure

Of the organic acids present in the Saskatoon berry, malic acid is the main representative (about 90%) [27]. Figure 1 and Figure 2 show the effect of the exposure time of gaseous ozone on the changes in pH and acidity of fruits of the Saskatoon berry.

The fruits of Saskatoon berries were characterized by an average acidity ranging from 0.276 g·100 g^−1^ (clone type H) to 1.508 g·100 g^−1^ (cultivar ‘Amela’) and a pH ranging from 4.12–5.03 (Figure 1 and Figure 2). In a study by Rop et al. [27], Saskatoon berry fruits were characterised by an acidity of 1.37–1.50 g·100 g^−1^, while the content of total organic acids in the study by Juríková et al. [28] amounted to 1.34–1.79 g·100 g^−1^. The ozonation process reduced the acidity (with the exception of the type H fruit) of Saskatoon berry by an average of 43.85% for fruit treated with 10 ppm gaseous ozone for 15 min and by an average of 26.39% for fruit treated with ozone for 30 min. Ozonation influenced the pH variation in the fruits of Saskatoon berry. Exposure to ozone for 15 min resulted in a pH increase of 5.32% on average (except for fruits of clone type H), while increasing the ozone exposure time by a further 15 min resulted in a pH increase of 6.33% on average for cultivar ‘Honeywood’ and clones type H and type N, and a pH decrease of 6.57% on average for cultivar ‘Thiessen’, ‘Amela’ and clone no 5/6 compared to fruits not treated with ozone (Figure 1 and Figure 2).

### 2.3. Influence of the Ozonation Process on the Sugar Profile of the Fruit of the Saskatoon Berry

The sugar content of Saskatoon berry and the ratio and proportion of its individual components (glucose, fructose, and sorbitol) have a significant impact on the perception of flavor intensity and consumer acceptance and determine the direction of its use and application in the food industry.

The fruits of the Saskatoon berries were characterized by a different sugar profile depending on the cultivar and clone (Table 3). Among the fruits that made up the control trial (no ozonation), the highest total sugar content (6.90 g·100 g^−1^ d.w.) and individual components (glucose and fructose) was characterized by clone type N, while the lowest content of total sugars in the non-ozonated fruits of the Saskatoon berry was characterized by clone type H (3.80 g·100 g^−1^ d.w.; Table 3). In a study by Lachowicz et al. [29], the sugar content (Σ glucose, fructose, and sorbitol) for clones no 5/6 and type N was significantly higher and amounted to, respectively: 9.11g·100 g^−1^ d.w. and 18.54 g·100 g^−1^ d.w., while in a study conducted in 2017, the total sugar content in the fruits of the ‘Honeywood’ cultivar was on average 16.01 g·100 g^−1^ d.w. [6]. The fruits of the N-type clone were characterized by the highest sugar content (Σ glucose, fructose, and sorbitol; 17.02 g·100 g^−1^ d.w. [6]. The sugar content in Saskatoon berry for type H and type N clones was also studied by Bieniek et al. [30]. The sugar content of the fruits analyzed was on a similar level and ranged from 11.30 g·100 g^−1^ d.w. for clone type N to 11.99 g·100 g^−1^ d.w. for clone type H. Both their own studies and those of the cited researchers’ teams were carried out on fruits of Saskatoon berry produced in particular years at the Orchard Institute in Skierniewice. The distribution of average temperatures and the sum of precipitation during the vegetation period of the Saskatoon berry, mainly in its final period (during fruit formation and ripening), significantly affects its chemical composition, including the content of sugars and their ratio and share of individual components (glucose, fructose and sorbitol). The content of nutritional compounds in the fruit depends on the degree of ripeness, growing and harvesting conditions, genotype and storage conditions [2,7]. The main dominant sugars in the fruits of the Saskatoon berries were glucose and fructose, which determine their taste qualities.

The ozonation process had a positive effect on the sugar profile of the ‘Thiessen’ fruit, as well as clones no 5/6 and type H, compared to the control sample, which was non-ozonated fruit. The ozonation process for 15 min had a significantly positive effect on the sugar content of cv. ‘Thiessen’ and type H, while extending the ozone exposure time to 30 min for clone no 5/6 (Table 3). Increasing the exposure time from 15 to 30 min resulted in a decrease in the sugar content of the ‘Thiessen’ cultivar and the type H clone fruit by an average of 9.31%. Ozonation for 15 min had a negative effect on sugar content in clone type N fruit, a decrease of 32.46% and in ‘Honeywood’ cultivar fruit—a decrease of 12.24% compared to the control. The conduct of the ozonation process in the cultivar and clone in question for 30 min resulted in a decrease in total sugar content by 26.14% on average, compared to fruits not subjected to ozonation (Table 3). In the ozonated fruits, the highest glucose content was characterized by the fruits of the Saskatoon berry cv. ‘Amela’ both after 15 min (3.20 g·100 g^−1^ d.w.) and after 30 min of ozonation (3.27 g·100 g^−1^ d.w.). The highest fructose content was found in the fruit of clone no 5/6 after 30 min of ozonation (2.03 g·100 g^−1^ d.w.).

### 2.4. The Content of Bioactive Compounds in the Fruit of Saskatoon Berry

Agronomic conditions, cultivar, harvest time, or storage and processing conditions can have a significant impact on the ascorbic acid content of the fruit [31]. Ascorbic acid is a chemical compound that has been shown to have positive effects in the treatment of scurvy or mucosal damage [31]. Furthermore, it strengthens the defense mechanisms of the human body [32]. Ascorbic acid is an antioxidant compound found in many fruits, for example, strawberry fruits contain an average of 65 mg·100 g^−1^, raspberry fruits contain an average of 29 mg·100 g^−1^, while blackberry fruits contain an average of 21 mg·100 g^−1^ ascorbic acid [31]. The non-ozonated fruits of the Saskatoon berry were characterized by an ascorbic acid content of 4.0–9.1 mg·100 g^−1^ (Figure 3.). In a study by Mazza [33], the fruits of Saskatoon berries were characterized by ascorbic acid contents of 3.05–4.38 mg·100 g^−1^. The ozonation process reduced the ascorbic acid content by an average of 20.74% for ozone of 15 min and by an average of 23.12% for ozone exposure of 30 min (Figure 3.)

The content of compounds with antioxidant activity contained in the Saskatoon berry mainly depends on its chemical composition, including the content of polyphenolic compounds and their differentiated structure affecting the antioxidant potential. The proportion of anthocyanins, procyanidins, phenolic compounds, triterpenes, or carotenoids in the Saskatoon berry significantly affects its antioxidant capacity [6]. It is also worth paying attention to agrotechnical, climatic conditions or plant genotype [29,33,34]. The antioxidant activity of the fruits of Saskatoon berry varied and ranged from 17.38 mM TE·100 g^−1^ d.w. for the cultivar ‘Honeywood’ to 21.04 mM TE·100 g^−1^ d.w. for the cultivar ‘Amela’ determined using the DPPH radical method and from 21.33 mM TE·100 g^−1^ d.w. for clone no 5/6 to 32.16 mM TE·100 g^−1^ d.w. for the cultivar ‘Amela’ determined using the ABTS cation radical (Table 4). The ozonation process increased the antioxidant activity of the Saskatoon berry as determined by the DPPH and ABTS methods, but these differences were not always statistically significant. The antioxidant potential of the fruits of Saskatoon berry determined by the ABTS method showed the highest activity (32.32 mM TE·100 g^−1^ d.w.) for cultivar ‘Thiessen’ while the lowest for clone no 5/6 (19.63 mM TE·100 g^−1^ d.w. [29]. While in a 2017 study, the average radical reduction capacity of Saskatoon berry fruit was 24.15 mM TE·100 g^−1^ d.w. [6].

Saskatoon berry fruit is a rich source of bioactive compounds that exhibit antioxidant, antiatherosclerotic, and antibacterial properties [35,36], mainly polyphenolic compounds, including anthocyanins and flavonoids [3,6]. The highest total polyphenol content among the fruits of Saskatoon berry not subjected to the ozonation process was characteristic for the ‘Thiessen’ cultivar, while the 15-min ozonation process caused a slight increase in the discussed compound by 8.71% on average (for the ‘Honeywood’ and ‘Amela’ cultivar). The 30-min ozonation process caused a significant increase in the total polyphenol content in the fruit of the ‘Amela’ cultivar (on average by 58.42% compared to the control. On the other hand, it had a significant negative effect on the total polyphenol content in the fruits of the Saskatoon berry ‘Thiessen’ cultivar (a decrease by an average of 42.44% compared to the control, which consisted of non-ozonated fruit; Figure 4).

The identification of polyphenolic compounds in the fruit of Saskatoon berry was based on the analysis of characteristic spectral data: mass-to-charge ratio *m/z* and radiant absorption maximum, which were compared with the available literature. Twenty-one polyphenolic compounds were identified, including four belonging to the anthocyanin group, the spectral properties of which are presented in Table 5. The ‘Thiessen’ cultivar had the highest anthocyanin content in the non-ozonated fruit of Saskatoon berry, with the cyanidin 3-O-glucoside content accounting for 28.90% of the total anthocyanin content. The fruits of the Saskatoon berry of the cultivar ‘Amela’ were characterized by the highest proportion of 3-O-galactoside cyanidin among the analyzed cultivars and clones (7014.80 mg·100 g^−1^ d.w.; Table 5). The ozonation process had a negative effect on the anthocyanin content of fruits of the Saskatoon berry; for the cultivars ‘Honeywood’ and ‘Thiessen’, the highest decrease was recorded after 15 min of the ozonation process (by 34.44% on average), while for the other cultivars and clones after 30 min of ozonation (by 57.18% on average; Table 5). In a study by Tian et al. [37], the anthocyanin content of the Saskatoon berry fruit was 222 mg·100 g^−1^ d.w., and the main compounds were cyanidin 3-O-galactoside, cyanidin 3-O-glucoside, and cyanidin 3-O-arabinoside. In a study by Lachowicz et al. [6], the average content of anthocyanins in the fruits of Saskatoon berry was 1504.9 mg·100 g^−1^ d.w., and the main chemical compound among the anthocyanins in the berries of the seven genotypes studied was cyanidin-3-O-galactoside (57% of all anthocyanins). In the study by Ribeiro de Souza et al. [38], the average anthocyanin content in the fruit of three varieties of Saskatoon berry was 171.57 mg·100 g^−1^ d.w., while in the study by Zhao et al. [3] the average anthocyanin content of the fruit was 55.6 mg·100 g^−1^ d.w., while in the study by Lachowicz et al. [2] it was slightly higher at 136.8 mg·100 g^−1^ d.w. The content of cyanidin 3-O-glucoside in the fruit of Saskatoon berry was 107.9 g·100 g^−1^ d.w. and was 67.47% higher compared to strawberry fruit, 38.65% higher compared to raspberry fruit, 7.6% higher compared to blueberry fruit, and 17.82% lower compared to chokeberry fruit [10]. The variation in the anthocyanin content of the fruits is not only related to the cultivar of the raw material used in the study, but also to the harvest date, the environmental conditions during the growing season of the plants, and the time and method of storage [8].

Of the other polyphenolic compounds contained in the non-ozonated and ozonated fruit of Saskatoon berry (although the ozonation process generally had a negative effect on the content of polyphenolic compounds), chlorogenic acid and quercetin 3-O-glucoside had the highest content (respectively, an average of 1855.33 mg·100 g^−1^ d.w. for non-ozonated fruit, an average of 1224.87 mg·100 g^−1^ d.w. for fruit ozonated for 15 min, and an average of 646.26 mg·100 g^−1^ d.w. for fruit ozonated for 30 min for chlorogenic acid content, and an average of 1297.09 mg·100 g^−1^ d.w. for non-ozonated fruit, an average of 943.42 mg·100 g^−1^ d.w. for fruit ozonated for 15 min and an average of 581.81 mg·100 g^−1^ d.w. for fruit ozonated for 30 min for quercetin 3-O-glucoside content; Table 5). Among Saskatoon berry cultivars, the type H clone was characterized by the highest content of the polyphenolic compounds discussed, both for ozonated and non-ozonated fruits (Table 5). The fruits of Saskatoon berries are characterized by a high content of phenolic acids, the main representatives of which are chlorogenic and neochlorogenic acids (the mean content was 59% and 18% of the total sum of phenolic acids in the Saskatoon berry, respectively; [6]). Phenolic acids are characterized by antioxidant activity, inhibit DNA structure and influence fruit flavour [6,39]. The average phenolic acid content in the seven analyzed varieties analyzed was 887.64 mg·100 g^−1^ d.w. [6]. The chlorogenic acid in the fruit of Saskatoon berry ranged from 86.62 to 298.13 mg·100 g^−1^ [28]. Flavone glycosides have potent antioxidant and anticancer properties and act as supportive agents in cardiovascular disease; in addition, they may be supportive substances in autoimmune diseases and for transplant patients [31]. Of the many identified flavonoid glycosides, the most notable are the glycosides of quercetin and kaempferol glycosides, which impart an astringent sensation in the mouth and, although to a much lower degree, also a bitter sensation [31]. In a study by Lachowicz et al. [6], the mean content of flavonols, whose representatives are derivatives of quercetin and kaempferol, was 288.87 mg·100 g^−1^ d.w., and the main flavonol compound in the seven varieties analyzed was quercetin-3-O-galactoside (approximately 60% of all flavonols). The average flavonol content in the study by Ribeiro de Souza et al. [38] was 74.27 mg·100 g^−1^ d.w. The quercetin content in the fruits of Saskatoon berry ranged from 8.44 to 30.68 mg·100 g^−1^ d.w. [28].

The fruits of the Saskatoon berry, especially the cultivar ‘Amela’, not subjected to ozonation, were also characterized by a relatively high content of (+) catechin—2112.92 mg·100 g^−1^ d.w., while the ozonation process reduced the (+) catechin content of (+) catechin by an average of 6.35% for the process lasting 15 min and by 88.1% for the process lasting 30 min (Table 5). (+) catechin is a chemical compound belonging to the flavone-3-ol group, which has a positive effect on the cardiovascular system by supporting proper regulation of blood flow [40]. In a study by Lachowicz et al. [6], the Saskatoon berry (+) catechin content of the fruit ranged from 17.59 to 38.05 mg·100 g^−1^ d.w. depending on the cultivar.

## 3. Materials and Methods

### 3.1. Material

Fruits of Saskatoon berries: two Canadian cultivars ‘Thiessen’ and ‘Honeywood’, one Polish cultivar ‘Amela’ and 3 Polish clones: clone no 5/6, clone type N, and clone type H constituted the research material. Ripe fruits weighing 1.5 kg each were harvested by hand from 6-year-old shrubs growing in an implementation experiment in the field at the Experimental Orchard in Dąbrowice (51.9163° N/20.1009° E) of the Institute of Horticulture—National Research Institute in Skierniewice, in the first decade of July 2021.

### 3.2. Treatment of Fruits by Ozone

The fruits of three cultivars and clones of Saskatoon berries were randomly divided into 500 g each. The first and second batches were ozoned using a Korona A40 Standard ozone generator (KoronaLab, Kolbudy, Poland) with an HRD OZ-1000 detector with a measurement range of 0–100 ppm. The first batch was treated with 10 ppm of ozone gas (flow time 4 m^3^·h^−1^, temperature 20 °C) for 15 min, while the second batch was treated with 10 ppm of ozone (flow time 4 m^3^·h^−1^, temperature 20 °C) for 30 min. The third batch was a control sample, not treated with ozonation. The ozonation process was carried out in triplicate. Ozonated and non-ozonated fruit samples were directly frozen and stored in a freezer (−18 °C) until physicochemical determinations were made.

### 3.3. Microbiological Analysis of the Saskatoon Berry

To determine the effectiveness of the ozonation process on the microbial load of Saskatoon berries, the ozonation treatment was repeated cyclically every 48 h. Fruit samples (200 g each) were stored in containers dedicated to food at 4 °C for 10 days. On 1st, 4th, 7th and 10th days, fruit samples were taken to determine the microbial load. The number of mesophilic aerobic bacteria and the number of yeasts and molds were determined according to the methodology described by Zardzewiały et al. [23].

### 3.4. PH and Acidity Determination

Total acidity of the Saskatoon berry fruit in terms of malic acid and pH were determined by potentiometric titration of the analyzed sample with a standard solution of 0.1 M NaOH until pH = 8.1 using a titrator (TitroLine 5000, Xylem Analytics, Weilheim, Germany) according to the method given in PN-EN 12147: 2000 [41]. All determinations were made in triplicate.

### 3.5. Determination of the Sugars Content

Briefly, 50 g of fruit samples were homogenized in a Unidrive X 1000 D, CAT homogenizer (with the addition of 100 mL of distilled water), followed by extraction using a Sonic-10 ultrasonic bath (15 min). The sugar content was measured by HPLC method with refractive index detection. The chromatographic equipment SYKAM (Eresing, Germany) consisting of sample injector S5250, pump system S1125, column oven S4120, and RI detector S3590 was used. Separation was carried out using a Cosmosil Sugar-D column (250 × 4.6 mm; Nacalai, San Diego, CA, USA). Separation was achieved with a mobile phase of 70% acetonitrile in water in isocratic mode. The flow rate was 0.5 mL/min at the column temperature set at 30 °C. The volume of the injected sample was 20 µL and 15 min was needed to complete the analysis. The samples before injection were centrifuged at 5000 rpm for 10 min. (Eppendorf Centrifuge 5430; Hamburg, Germany) and diluted with mobile phase 1:4 (*v*/*v*). All determinations were made in triplicate.

### 3.6. Determination of Bioactive Components

The determination of the ascorbic acid content in the Saskatoon berry was carried out according to PN-A-04019:1998 [42]. The content of total polyphenols in the fruits of the Saskatoon berry was determined using the Folin-Ciocalteu method according to the methodology described by Piechowiak et al. [43], whereas the identification of individual polyphenolic compounds was carried out using the UPLC equipped with a binary pump, column and sample manager, photodiode array detector (PDA), tandem quadrupole mass spectrometer (TQD) with electrospray ionization (ESI) source working in negative mode (Waters, Milford, MA, USA) according to the method of Żurek et al. [44]. Separation was carried out using the UPLC BEH C18 column (1.7 µm, 100 mm × 2.1 mm, Waters) at 50 °C, with a constant flow rate of 0.35 mL/min. The injection volume of the samples was 5 µL. The eluent was a mixture of water (solvent A) and 40% acetonitrile in water, *v*/*v* (solvent B). The TQD parameters were as follows: capillary voltage of 3500 V; con voltage of 30 V; con gas flow 100 L/h; source temperature 120 °C; desolvation temperature 350 °C; and desolvation gas flow rate of 800 L/h. Polyphenolic identification was performed on the basis of the mass-to-charge ratio, retention time, specific PDA spectra, fragment ions, and comparison of data obtained with and literature findings and commercial standards. The results were expressed as mg·100 g^−1^ d.w. The antioxidant activity of DPPH· and ABTS^+^ was assessed according to the methodology described by Piechowiak et al. [44]. All analyzes were performed in triplicate.

### 3.7. Statistical Analysis

Using the Statistica program 13.3. (TIBCO Software Inc., Tulsa, OK, USA), a statistical analysis of the results obtained was performed, including a multivariate analysis of variance (ANOVA) and a significance test at a significance level of α = 0.05 by comparing the results between the variants of the experiment.

## 4. Conclusions

The study concluded that the ozonation process had a positive effect in reducing the microbial load of Saskatoon berry fruit on the 7th day of cold storage, including a significant reduction in aerobic mesophilic bacteria, yeast, and molds and a decrease in fruit acidity, especially for an ozone concentration of 10 ppm and an exposure time of 30 min, compared to the control sample. Exposure of Saskatoon berry fruit to gaseous ozone resulted in a variation in sugar content mainly for clone no 5/6 after 30 min of ozonation. Among the cultivars and clones of Saskatoon berry, clone type H was characterized by the highest content of polyphenolic compounds, both ozonated and non-ozonated fruits. The content of ascorbic acid and antioxidant activity (determined by DPPH· and ABTS^+^ methods) varied according to the cultivar and clone tested as well as the ozone exposure time.

## Figures and Tables

**Figure 1 molecules-27-06446-f001:**
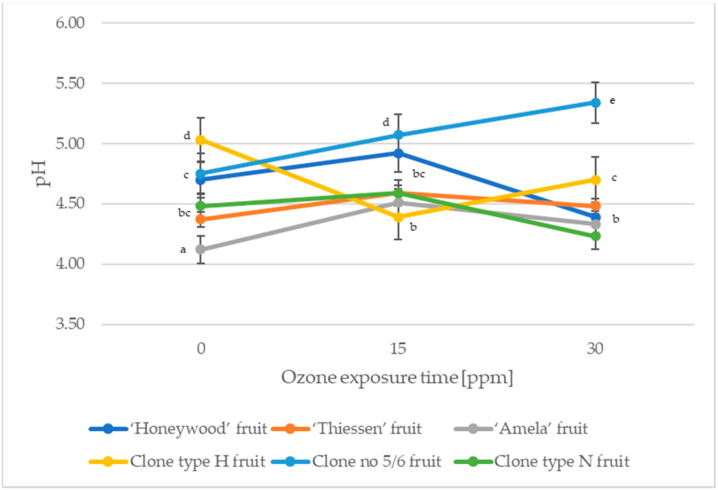
Changes in the pH of the Saskatoon berry during ozone exposure. Data are expressed as mean values (n = 3) ± SD; SD—standard deviation. Mean values with different letters are significantly different (*p* < 0.05).

**Figure 2 molecules-27-06446-f002:**
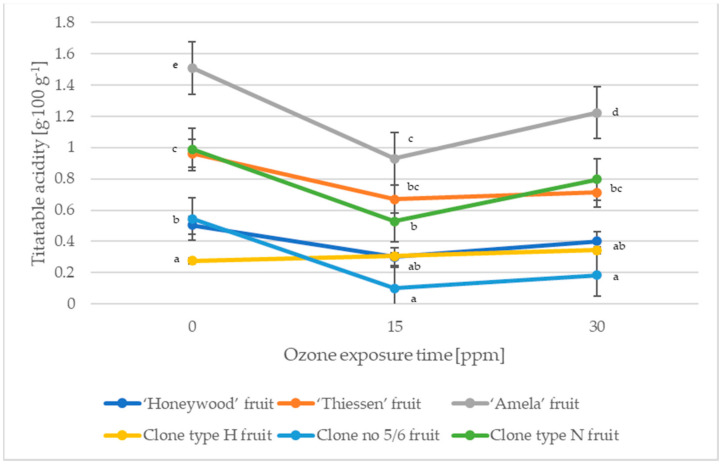
Changes in the acidity of the Saskatoon berry fruit during ozone exposure. Data are expressed as mean values (n = 3) ± SD; SD—standard deviation. Mean values with different letters are significantly different (*p* < 0.05).

**Figure 3 molecules-27-06446-f003:**
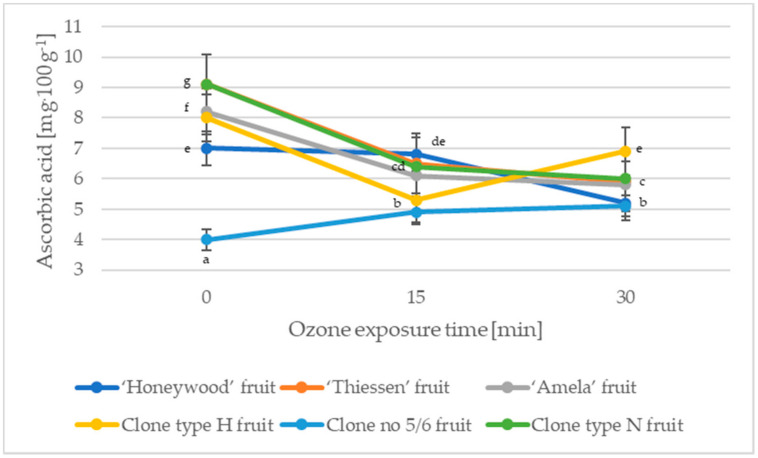
Changes in the ascorbic acid of the Saskatoon berry fruit during ozone exposure. Data are expressed as mean values (n = 3) ± SD; SD—standard deviation. Mean values with different letters are significantly different (*p* < 0.05).

**Figure 4 molecules-27-06446-f004:**
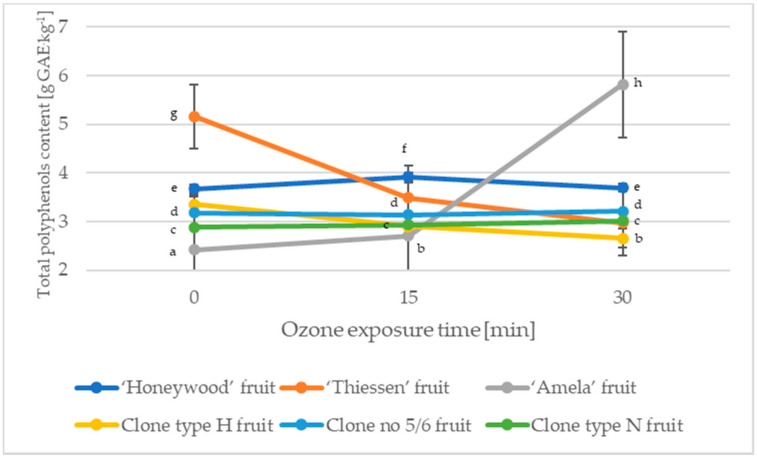
Changes in the total polyphenolic content of Saskatoon berry fruit during ozone. Data are expressed as mean values (n = 3) ± SD; SD—standard deviation. Mean values with different letters are significantly different (*p* < 0.05).

**Table 1 molecules-27-06446-t001:** Microbiological load of mesophilic aerobic bacteria in the fruit of Saskatoon berry during storage.

Fruit Cultivar	Ozone Exposure Time [min]	The Date of the Test			
		1 Day after Ozonation [log cfu g^−1^]	4 Days after Ozonation [log cfu g^−1^]	7 Days after Ozonation [log cfu g^−1^]	10 Days after Ozonation [log cfu g^−1^]
	0	6.32 ^aA^	6.41^bA^	6.51 ^bB^	6.56 ^bB^
‘Honeywood’	15	6.20 ^aA^	6.28 ^aA^	6.04 ^aA^	6.32 ^aA^
	30	6.18 ^aA^	6.15 ^aA^	6.01 ^aA^	6.20 ^aA^
	0	6.51 ^bB^	6.62 ^cC^	6.65 ^cC^	6.71 ^cC^
‘Thiessen’	15	6.48 ^bB^	6.45 ^aB^	6.34 ^aA^	6.41 ^aA^
	30	6.46 ^bB^	6.34 ^bB^	6.20 ^aA^	6.29 ^aA^
	0	6.34 ^aA^	6.36 ^aA^	6.46 ^bA^	6.54 ^bB^
‘Amela’	15	6.32 ^aA^	6.31 ^bA^	6.23 ^aA^	6.26 ^aA^
	30	6.23 ^aA^	6.15 ^aA^	6.08 ^aA^	6.18 ^aA^
	0	6.40 ^aB^	6.43 ^bB^	6.49 ^bC^	6.54 ^bC^
Clone type H	15	6.34 ^aB^	6.32 ^aB^	6.26 ^aA^	6.28 ^aA^
	30	6.28 ^aA^	6.15 ^aA^	6.08 ^aA^	6.15 ^aA^
	0	6.45 ^aA^	6.52 ^bA^	6.5 ^cB^	6.65 ^cC^
Clone no 5/6	15	6.42 ^aA^	6.38 ^bA^	6.32 ^aA^	6.38 ^aA^
	30	6.41 ^aA^	6.34 ^bA^	6.28 ^aA^	6.34 ^aA^
	0	6.18 ^aA^	6.34 ^bB^	6.43 ^bB^	6.52 ^bC^
Clone type N	15	6.17 ^aA^	6.15 ^aA^	6.08 ^aA^	6.23 ^aA^
	30	6.16 ^aA^	6.11 ^aA^	6.04 ^aA^	6.15 ^aA^

Data are expressed as mean values (n = 3). Different small letters denote differences in the results between ozone doses on individual days, and different capital letters indicate differences between the measurements dates, *p* < 0.05.

**Table 2 molecules-27-06446-t002:** Microbial load from yeasts and molds of Saskatoon berry fruit during storage.

Fruit Cultivar	Ozone Exposure Time [min]	The Date of the Test			
		1 Day after Ozonation [log cfu g^−1^]	4 Days after Ozonation [log cfu g^−1^]	7 Days after Ozonation [log cfu g^−1^]	10 Days after Ozonation [log cfu g^−1^]
	0	3.93 ^cC^	3.94 ^cC^	4.02 ^cC^	4.15 ^bC^
‘Honeywood’	15	3.63 ^aA^	3.61 ^aA^	3.57 ^aA^	3.61 ^aA^
	30	3.61 ^aA^	3.50 ^aA^	3.36 ^aA^	3.38 ^aA^
	0	3.96 ^cC^	4.03 ^cC^	4.11 ^cC^	4.22 ^cC^
‘Thiessen’	15	3.71 ^aA^	3.70 ^aA^	3.69 ^aA^	3.72 ^aA^
	30	3.72 ^aA^	3.72 ^aA^	3.56 ^aA^	3.71 ^aA^
	0	3.88 ^bB^	3.90 ^bB^	3.96 ^bB^	4.02 ^bC^
‘Amela’	15	3.51 ^aA^	3.49 ^aA^	3.48 ^aA^	3.51 ^aA^
	30	3.45 ^aA^	3.46 ^aA^	3.45 ^aA^	3.46 ^aA^
	0	3.84 ^bB^	3.94 ^bB^	4.01 ^bB^	4.06 ^bB^
Clone type H	15	3.54 ^aA^	3.51 ^aA^	3.49 ^aA^	3.52 ^aA^
	30	3.53 ^aA^	3.52 ^aA^	3.54 ^aA^	3.51 ^aA^
	0	3.81 ^bB^	3.83 ^bB^	3.85 ^bB^	3.95 ^bB^
Clone no 5/6	15	3.49 ^aA^	3.51 ^aA^	3.4 8^aA^	3.54 ^aA^
	30	3.48 ^aA^	3.52 ^aA^	3.47 ^aA^	3.51 ^aA^
	0	3.97 ^cC^	3.96 ^cC^	4.01 ^cC^	4.04 ^cC^
Clone type N	15	3.62 ^aA^	3.63 ^aA^	3.60 ^aA^	3.62 ^aA^
	30	3.60 ^aA^	3.53 ^aA^	3.59 ^aA^	3.60 ^aA^

Data are expressed as mean values (n = 3). Different small letters denote differences in the results between ozone doses on individual days, and different capital letters indicate differences between the measurements dates, *p* < 0.05.

**Table 3 molecules-27-06446-t003:** The sugar profile of the fruit of the Saskatoon berry.

Fruit Cultivar	Ozone Exposure Time [min]	Glucose [g·100 g^−1^ d. w.]	Fructose [g·100 g^−1^ d. w.]	Sorbitol [g·100 g^−1^ d. w.]	Σ Sugar
	0	3.02 ^fg^ ± 0.11	1.63 ^cd^ ± 0.15	1.39 ^gh^ ± 0.10	6.21 ^h^ ± 0.09
‘Honeywood’	15	2.72 ^de^ ± 0.10	1.44 ^abc^ ± 0.18	1.28 ^fg^ ± 0.11	5.45 ^fg^ ± 0.08
	30	2.21 ^bc^ ± 0.14	1.20 ^a^ ± 0.15	1.01 ^cde^ ± 0.20	4.43 ^c^ ± 0.15
	0	2.17 ^b^ ± 0.11	1.54 ^bc^ ± 0.15	0.64 ^a^ ± 0.14	4.34 ^c^ ± 0.17
‘Thiessen’	15	2.56 ^de^ ± 0.21	1.70 ^cd^ ± 0.10	0.86 ^bcd^ ± 0.13	5.11 ^e^ ± 0.10
	30	2.32 ^c^ ± 0.13	1.52 ^bc^ ± 0.19	0.80 ^ab^ ± 0.10	4.65 ^d^ ± 0.05
	0	3.30 ^hi^ ± 0.08	1.70 ^cd^ ± 0.07	1.60 ^hi^ ± 0.13	6.60 ^i^ ± 0.08
‘Amela’	15	3.20 ^gh^ ± 0.18	1.62 ^cd^ ± 0.14	1.58 ^hi^ ± 0.11	6.40 ^hi^ ± 0.15
	30	3.27 ^hi^ ± 0.19	1.61 ^cd^ ± 0.13	1.66 ^i^ ± 0.08	6.53 ^i^ ± 0.14
	0	1.90 ^a^ ± 0.18	1.20 ^a^ ± 0.15	0.70 ^a^ ± 0.22	3.80 ^a^ ± 0.15
Clone type H	15	2.24 ^bc^ ± 0.15	1.41 ^a^ ± 0.19	0.83 ^abc^ ± 0.13	4.47 ^cd^ ± 0.12
	30	2.02 ^ab^ ± 0.13	1.25 ^a^ ± 0.05	0.77 ^ab^ ± 0.11	4.04 ^b^ ± 0.17
	0	2.80 ^ef^ ± 0.13	1.81 ^de^ ± 0.18	0.98 ^bcde^ ± 0.08	5.60 ^g^ ± 0.12
Clone no 5/6	15	2.83 ^ef^ ± 0.17	1.91 ^def^ ± 0.10	0.92 ^bcd^ ± 0.08	5.07 ^e^ ± 0.11
	30	3.14 ^gh^ ± 0.08	2.03 ^ef^ ± 0.16	1.11 ^efg^ ± 0.11	6.28 ^h^ ± 0.11
	0	3.45 ^i^ ± 0.15	2.14 ^f^ ± 0.06	1.31 ^g^ ± 0.21	6.90 ^j^ ± 0.12
Clone type N	15	2.33 ^c^ ± 0.07	1.34 ^ab^ ± 0.24	0.99 ^cde^ ± 0.10	4.66 ^d^ ± 0.06
	30	2.64 ^de^ ± 0.04	1.56 ^c^ ± 0.15	1.07 ^def^ ± 0.14	5.27 ^ef^ ± 0.14

Data are expressed as mean values (n = 3) ± SD; SD—standard deviation. Mean values within columns with different letters are significantly different (*p* < 0.05).

**Table 4 molecules-27-06446-t004:** The content of the antioxidant activity of the Saskatoon berry.

Fruit Cultivar	Ozone Exposure Time [min]	Antioxidant Activity	
		DPPH [mM TE·100g^−1^ d.w.]	ABTS^+^ [mM TE·100g^−1^ d.w.]
	0	17.38 ^ab^ ± 0.10	31.20 ^j^ ± 0.09
‘Honeywood’	15	17.59 ^b^ ± 0.26	31.73 ^k^ ± 0.26
	30	17.21 ^a^ ± 0.12	32.06 ^l^ ± 0.28
	0	20.97 ^fg^ ± 0.13	30.06 ^h^ ± 0.22
‘Thiessen’	15	21.07 ^fg^ ± 0.16	29.97 ^h^ ± 0.15
	30	21.16 ^g^ ± 0.16	31.13 ^j^ ± 0.09
	0	21.04 ^fg^ ± 0.04	32.16 ^l^ ± 0.16
‘Amela’	15	21.16 ^g^ ± 0.06	31.07 ^j^ ± 0.07
	30	20.88 ^fg^ ± 0.13	32.53 ^m^ ± 0.03
	0	18.99 ^cd^ ± 0.26	25.18 ^f^ ± 0.18
Clone type H	15	19.44 ^de^ ± 0.11	24.75 ^e^ ± 0.10
	30	19.67 ^e^ ± 0.09	22.67 ^d^ ± 0.16
	0	20.88 ^fg^ ± 0.23	21.33 ^b^ ± 0.24
Clone no 5/6	15	21.14 ^g^ ± 0.14	21.87 ^c^ ± 0.13
	30	21.42 ^h^ ± 0.04	20.88 ^a^ ± 0.13
	0	18.95 ^c^ ± 0.05	29.33 ^g^ ± 0.16
Clone type N	15	19.21 ^d^ ± 0.12	29.96 ^h^ ± 0.04
	30	19.50 ^e^ ± 0.30	30.43 ^i^ ± 0.10

Data are expressed as mean values (n = 3) ± SD; SD—standard deviation. Mean values within columns with different letters are significantly different (*p* < 0.05).

**Table 5 molecules-27-06446-t005:** Individual phenolic compounds identified by UPLC-PDA-MS/MS in Saskatoon berries.

Compound		Rt	λ_max_	[M-H] m/z		‘Honeywood’			‘Thiessen’			‘Amela’		
[mg·100 g^−1^ d.w.]		(min)	nm	MS	MS/MS	0	15	30	0	15	30	0	15	30
Anthocyanins														
1	Cyanidin 3-O-galactoside	2.52	279, 515	449+	287	6119.91 ^g^ ± 1.12	736.76 ^ab^ ± 0.98	1447.30 ^b^ ± 2.13	6788.96 ^h^ ± 1.87	1630.43 ^b^ ± 1.01	2032.50 ^c^ ± 0.89	7014.80 ^h^ ± 1.12	6832.15 ^h^ ± 2.74	1869.40 ^bc^ ± 1.74
2	Cyanidin 3-O-glucoside	2.76	279, 514	449+	287	3485.43 ^i^ ± 2.55	259.63 ^b^ ± 1.76	620.68 ^cd^ ± 2.01	3942.85 ^j^ ± 3.16	754.76 ^de^ ± 1.22	926.08 ^ef^ ± 2.65	1031.46 ^f^ ± 3.41	997.11 ^f^ ± 2.98	245.17 ^ab^ ± 1.75
3	Cyanidin 3-O-arabinoside	2.99	279, 515	419+	287	1388.30 ^i^ ± 3.02	106.32 ^a^ ± 2.37	264.89 ^bc^ ± 1.22	1664.07 ^j^ ± 1.54	300.72 ^c^ ± 1.76	372.98 ^c^ ± 2.39	862.04 ^g^ ± 2.13	726.57 ^f^ ± 2.38	162.84 ^ab^ ± 1.49
4	Cyanidin 3-O-xyloside	3.36	281, 517	419+	287	1062.72 ^g^ ± 2.51	75.41 ^a^ ± 2.73	180.59 ^c^ ± 2.69	1245.41 ^h^ ± 2.90	207.04 ^cd^ ± 1.65	264.46 ^cd^ ± 2.08	503.94 ^e^ ± 1.78	433.15 ^e^ ± 2.86	101.74 ^b^ ± 1.64
Other phenolics														
5	Neochlorogenic acid	2.24	288sh, 324	353-	191	421.75 ^h^ ± 1.99	20.15 ^a^ ± 1.33	43.83 ^ab^ ± 1.84	372.00 ^g^ ± 3.05	38.13 ^ab^ ± 3.00	64.79 ^b^ ± 2.68	360.83 ^g^ ± 1.65	347.93 ^g^ ± 2.66	56.26 ^abc^ ± 2.44
6	Chlorogenic acid	2.85	288sh, 324	353-	191	2377.23 ^f^±	127.80 ^a^ ± 1.22	292.13 ^ab^ ± 1.49	2370.38 ^f^ ± 1.69	266.57 ^ab^ ± 1.41	451.51 ^bc^ ± 1.51	1948.70 ^e^ ± 1.62	1799.27 ^e^ ± 1.72	295.07^ab^ ± 1.49
7	Procyanidin dimer B-type	3.01	279	577-	289	563.80 ^g^ ± 1.56	47.59 ^a^ ± 2.56	101.84 ^b^ ± 1.22	545.94 ^g^ ± 2.16	113.21 ^b^ ± 1.35	132.78 ^b^ ± 2.55	628.94 ^g^ ± 1.55	592.16 ^g^ ± 1.55	148.05 ^bc^ ± 2.32
8	Coumaric acid glucoside	3.06	309	325-	163	271.08 ^g^ ± 2.31	18.82 ^a^ ± 2.17	39.76 ^bcd^ ± 1.59	279.97 ^g^ ± 2.88	51.21 ^cd^ ± 1.69	63.95 ^de^ ± 2,83	72.35 ^e^ ± 2.01	74.79 ^e^ ± 1.69	18.89 ^ab^ ± 2.11
9	Procyanidin dimer B-type	3.25	279	577-	289	68.82 ^f^ ± 3.19	5.00 ^a^ ± 2.89	13.34 ^a^ ± 1.85	114.04 ^g^ ± 2.64	14.29 ^ab^ ± 1.78	21.42 ^bc^ ± 1.35	29.68 ^c^ ± 1.89	31.61 ^c^ ± 1.89	12.93 ^a^ ± 1.98
10	(+) Catechin	3.40	274	289-	141	683.61 ^c^ ± 1.45	56.62 ^b^ ± 1.34	92.61 ^bc^ ± 1.79	572.86 ^c^ ± 2.34	71.77 ^b^ ± 1.94	69.41 ^b^ ± 1.53	2112.92 ^g^ ± 2.61	1978.65 ^g^ ± 2.14	251.59 ^c^ ± 2.36
11	Coumaroilo-quinic acid	3.56	308	337-	163	124.37 ^c^ ± 1.66	24.80 ^a^ ± 1.67	47.54 ^a^ ± 1.99	215.36 ^d^ ± 2.99	53.99 ^ab^ ± 1.66	55.53 ^ab^ ± 1.46	127.81 ^c^ ± 2.17	154.53 ^c^ ± 2.38	36.40 ^a^ ± 2.51
12	Coutaric acid	3.79	311	295-	163	302.34 ^g^ ± 1.97	15.12 ^a^ ± 1.54	37.71 ^b^ ± 2.37	298.33 ^g^ ± 3.16	25.51 ^ab^ ± 2.13	38.77 ^b^ ± 1.82	400.31 ^i^ ± 1.63	340.39 ^h^ ± 1.98	62.17 ^c^ ± 2.41
13	Quercetin 3-O-arabinoside-glucoside	4.21	255, 355	595-	301	263.44 ^d^ ± 2.18	42.00 ^a^ ± 3.06	75.77 ^a^ ± 3.18	338.77 ^e^ ± 3.47	59.18 ^a^ ± 1.56	72.19 ^a^ ± 2.49	795.50 ^g^ ± 1.32	796.85 ^g^ ± 2.13	177.66 ^bc^ ± 1.88
14	Quercetin 3-O-rutinoside	4.41	255, 355	609-	301	460.00 ^h^ ± 2.26	60.04 ^a^ ± 2.49	97.68 ^ab^ ± 2.46	453.31 ^h^ ± 2.58	111.98 ^bc^ ± 1.46	122.54 ^c^ ± 2.13	427.08 ^gh^ ± 1.48	433.46 ^gh^ ± 2.41	135.12 ^cd^ ± 1.41
15	Kaempferol 3-O-glucuronide	4.49	264, 338	461-	285	116.05 ^g^ ± 2.49	12.18 ^a^ ± 1.74	18.77 ^a^ ± 1.10	125.58 ^gh^ ± 2.13	23.04 ^ab^ ± 2.85	26.55 ^b^ ± 2.31	108.58 ^f^ ± 1.52	113.00 ^fg^ ± 2.69	27.98 ^b^ ± 1.63
16	Quercetin 3-O-glucoside	4.57	255, 355	463-	301	1787.98 ^h^ ± 1.18	271.72 ^ab^ ± 1.39	459.53 ^c^ ± 1.56	1941.97 ^i^ ± 2.00	549.81 ^d^ ± 2.16	525.40 ^d^ ± 2.47	1378.46 ^g^ ± 2.01	1418.09 ^g^ ± 2.36	454.05 ^cd^ ± 1.52
17	Quercetin 3-O-galactoside	4.71	255, 355	463-	301	259.56 ^e^ ± 1.74	26.75 ^a^ ± 2.12	53.87 ^bc^ ± 1.87	283.87 ^f^ ± 2.01	52.94 ^bc^ ± 2.48	44.81 ^ab^ ± 1.74	263.20 ^ef^ ± 2.02	257.90 ^e^ ± 2.46	62.6 1^bc^ ± 2.16
18	Quercetin 3-O-pentoside I	4.93	255, 355	433-	301	74.98 ^e^ ± 1.64	9.54 ^a^ ± 1.49	18.88 ^ab^ ± 1.74	73.25 ^e^ ± 1.67	19.66 ^ab^ ± 1.15	19.32 ^ab^ ± 2.83	31.51 ^c^ ± 1.79	29.30 ^bc^ ± 2.72	11.26 ^a^ ± 2.55
19	Quercetin 3-O-pentoside II	5.01	255, 355	433-	301	156.32 ^g^ ± 3.08	24.91 ^a^ ± 2.56	45.38 ^bc^ ± 1.41	152.68 ^g^ ± 1.49	38.50 ^b^ ± 1.28	40.12 ^b^ ± 2.71	74.83 ^d^ ± 1.06	77.41 ^d^ ± 1.83	23.97 ^a^ ± 2.01
20	Quercetin 3-O-pentoside III	5.11	255, 355	433-	301	41.55 ^e^ ± 2.18	11.06 ^ab^ ± 2.81	17.49 ^bc^ ± 2.67	37.45 ^e^ ± 2.55	10.50 ^ab^ ± 1.49	16.29 ^b^ ± 2.01	20.24 ^c^ ± 2.54	20.12 ^c^ ± 1.61	10.28 ^ab^ ± 2.61
21	3,4-di-O-caffeoyl-quinic acid	5.20	288sh, 324	515-	353	221.33 ^ij^ ± 1.19	21.80 ^a^ ± 2.39	42.35 ^b^ ± 2.61	231.46 ^j^ ± 2.49	31.66 ^ab^ ± 1.75	47.04 ^bc^ ± 1.16	209.93 ^hi^ ± 2.42	198.25 ^h^ ± 1.78	42.50 ^b^ ± 1.17
Total polyphenols [g·100 g^−1^ d.w.]						20.25 ^k^ ± 2.13	1.97 ^a^ ± 1.17	4.01 ^cd^ ± 1.17	22.05 ^k^ ± 1.64	4.42 ^d^ ± 2.12	5.41 ^de^ ± 1.99	18.40 ^ij^ ± 1.69	17.65 ^i^ ± 2.22	4.21 ^cd^ ± 1.26
**Compound**		**Rt**	**λ_max_**	**[M-H] m/z**		**Clone Typ H**			**Clone No. 5/6**			**Clone Typ N**		
**[mg·100 g^−1^ d.w.]**		**(min.)**	**nm**	**MS**	**MS/MS**	**0**	**15**	**30**	**0**	**15**	**30**	**0**	**15**	**30**
Anthocyanins														
1	Cyanidin 3-O-galactoside	2.52	279, 515	449+	287	6613.36 ^gh^ ± 1.79	5857.53 ^fg^ ± 3.21	5277.88 ^f^ ± 2.12	1259.10 ^b^ ± 2.87	3701.10 ^d^ ± 3.14	611.91 ^a^ ± 1.09	2206.33 ^c^ ± 2.18	4447.50 ^e^ ± 2.98	1658.29 ^b^ ± 3.01
2	Cyanidin 3-O-glucoside	2.76	279, 514	449+	287	2271.37 ^h^ ± 2.45	2284.64 ^h^ ± 2.69	2060.35 ^g^ ± 1.13	195.39 ^a^ ± 1.46	542.01 ^c^ ± 1.98	84.88 ^a^ ± 2.09	351.78 ^b^ ± 1.34	865.41 ^ef^ ± 2.37	240.92 ^ab^ ± 1.78
3	Cyanidin 3-O-arabinoside	2.99	279, 515	419+	287	1067.73 ^h^ ± 3.97	993.01 ^h^ ± 3.23	979.21 ^h^ ± 1.67	141.73 ^ab^ ± 1.51	396.57 ^c^ ± 1.87	59.32 ^a^ ± 1.11	244.25 ^bc^ ± 0.97	557.68 ^d^ ± 2.08	165.78 ^ab^ ± 2.31
4	Cyanidin 3-O-xyloside	3.36	281, 517	419+	287	801.68 ^f^ ± 1.99	799.17 ^f^ ± 2.06	734.74 ^f^ ± 2.33	73.39 ^a^ ± 1.78	200.10 ^cd^ ± 2.55	30.40 ^a^ ± 1.88	150.17 ^bc^ ± 1.52	343.67 ^d^ ± 2.00	95.86 ^ab^ ± 1.82
Other phenolics														
5	Neochlorogenic acid	2.24	288sh, 324	353-	191	549.99 ± 2.08	451.57 ^h^ ± 2.64	345.00 ^g^ ± 1.22	95.67 ^cd^ ± 1.45	293,49 ^f^ ± 2.77	48.30 ^ab^ ± 1.89	115.50 ^d^ ± 1.55	177.27 ^e^ ± 1.70	61.17 ^bc^ ± 1.59
6	Chlorogenic acid	2.85	288sh, 324	353-	191	3458.52 ^h^ ± 3.03	3014.45 ^g^ ± 2.18	2336.02 ^f^ ± 1.22	360.19 ^b^ ± 1.17	1111.06 ^d^ ± 3.17	164.79 ^a^ ± 1.52	616.93 ^c^ ± 1.51	1030.07 ^d^ ± 1.33	338.05 ^b^ ± 2.47
7	Procyanidin dimer B-type	3.01	279	577-	289	421.25 ^f^ ± 3.12	359.94 ^ef^ ± 2.65	378.50 ^ef^ ± 2.36	86.99 ^ab^ ± 1.65	258.59 ^d^ ± 2.85	34.04 ^a^ ± 3.01	207.10 ^cd^ ± 1.26	316.64 ^de^ ± 1.18	148.96 ^bc^ ± 2.94
8	Coumaric acid glucoside	3.06	309	325-	163	136.53 ^f^ ± 2.89	144.75 ^f^ ± 1.45	135.00 ^f^ ± 2.49	11.06 ^ab^ ± 1.87	31.09 ^bc^ ± 2.36	5.16 ^a^ ± 2.62	27.03 ^bc^ ± 1.69	40.06 ^bcd^ ± 2.11	19.74 ^ab^ ± 1.26
9	Procyanidin dimer B-type	3.25	279	577-	289	54.27 ^de^ ± 2.11	44.89 ^d^ ± 1.22	58.92 ^e^ ± 2.74	6.26 ^a^ ± 1.13	15.39 ^ab^ ± 2.51	9.50 ^a^ ± 1.23	10.76 ^a^ ± 1.48	25.37 ^bc^ ± 2.71	13.65 ^ab^ ± 1.65
10	(+) Catechin	3.40	274	289-	141	1597.33 ^f^ ± 2.19	1206.91 ^e^ ± 1.18	1018.92 ^d^ ± 2.61	24.37 ^b^ ± 1.27	55.54 ^b^ ± 2.81	6.93 ^a^ ± 1.05	618.81 ^c^ ± 1.12	948.91 ^d^ ± 2.94	265.09 ^c^ ± 1.39
11	Coumaroilo-quinic acid	3.56	308	337-	163	293.64 ^e^ ± 2.17	272.08 ^e^ ± 2.18	236.00 ^de^ ± 2.91	49.32 ^a^ ± 1.67	156.36 ^c^ ± 1.64	29.89 ^a^ ± 1.26	50.96 ^a^ ± 1.64	78.70 ^b^ ± 2.33	44.91 ^a^ ± 1.37
12	Coutaric acid	3.79	311	295-	163	158.98 ^e^ ± 3.13	137.55 ^e^ ± 1.69	105.11 ^d^ ± 1.91	21.84 ^ab^ ± 1.97	55.29 ^c^ ± 1.12	9.58 ^a^ ± 1.31	107.03 ^d^ ± 1.66	194.90 ^f^ ± 1.41	50.81 ^bc^ ± 1.88
13	Quercetin 3-O-arabinoside-glucoside	4.21	255, 355	595-	301	429.95 ^f^ ± 1.51	402.50 ^f^ ± 1.96	324.44 ^e^ ± 1.39	137.02 ^b^ ± 1.74	359.51 ^e^ ± 1.43	70.46 ^a^ ± 2.16	288.43 ^d^ ± 2.12	379.26 ^e^ ± 1.56	186.93 ^c^ ± 2.91
14	Quercetin 3-O-rutinoside	4.41	255, 355	609-	301	321.12 ^f^ ± 1.45	324.24 ^f^±	251.07 ^e^ ± 1.55	147.19 ^cd^ ± 2.12	407.68 ^g^ ± 1.36	80.99 ^ab^ ± 2.41	170.64 ^d^ ± 1.66	219.35 ^e^ ± 1.82	120.07 ^c^ ± 2.73
15	Kaempferol 3-O-glucuronide	4.49	264, 338	461-	285	134.83 ^h^ ± 1.69	124.06 ^gh^ ± 2.18	93.74 ^e^ ± 1.69	22.97 ^ab^ ± 2.41	67.25 ^d^ ± 1.84	10.84 ^a^ ± 2.51	42.30 ^c^ ± 1.15	56.16 ^d^ ± 1.16	27.02 ^b^ ± 2.41
16	Quercetin 3-O-glucoside	4.57	255, 355	463-	301	1760.06 ^h^ ± 1.92	1691.82 ^h^ ± 2.46	1378.77 ^g^ ± 1.12	358.94 ^bcd^ ± 2.36	1018.56 ^f^ ± 2.51	203.50 ^a^ ± 2.73	555.11 ^d^ ± 1.05	710.52 ^e^ ± 1.83	469.63 ^cd^ ± 2.59
17	Quercetin 3-O-galactoside	4.71	255, 355	463-	301	400.75 ^h^ ± 1.78	372.05 ^g^ ± 1.36	297.71 ^f^ ± 1.08	32.59 ^ab^ ± 2.91	107.60 ^d^ ± 2.37	21.55 ^a^ ± 2.34	104.56 ^d^ ± 1.09	124.56 ^d^ ± 1.33	70.33 ^c^ ± 2.11
18	Quercetin 3-O-pentoside I	4.93	255, 355	433-	301	67.11 ^e^ ± 2.39	72.81 ^e^ ± 1.73	56.54 ^d^ ± 2.78	24.51 ^bc^ ± 2.41	68.29 ^e^ ± 2.51	16.07 ^a^ ± 1.59	12.74 ^a^ ± 1.39	20.66 ^b^ ± 1.61	12.69 ^a^ ± 1.13
19	Quercetin 3-O-pentoside II	5.01	255, 355	433-	301	126.31 ^f^ ± 1.79	133.88 ^f^ ± 1.18	105.91 ^e^ ± 2.54	23.68 ^a^ ± 1.74	62.01 ^cd^ ± 2.71	14.34 ^a^ ± 1.74	32.94 ^b^ ± 1.51	37.48 ^b^ ± 1.28	30.40 ^ab^ ± 1.76
20	Quercetin 3-O-pentoside III	5.11	255, 355	433-	301	98.79 ^g^ ± 1.09	101.42 ^g^ ± 2.13	86.37 ^f^ ± 2.06	27.65 ^d^ ± 1.94	101.17 ^g^ ± 1.64	22.73 ^cd^ ± 1.33	6.81 ^a^ ± 1.84	9.01 ^ab^ ± 2.14	9.49 ^ab^ ± 1.55
21	3,4-di-O-caffeoyl-quinic acid	5.20	288sh, 324	515-	353	79.62 ^e^ ± 3.16	70.62 ^de^ ± 1.79	64.13 ^d^ ± 1.16	54.44 ^cd^ ± 1.52	138.40 ^g^ ± 1.32	26.26 ^a^ ± 1.84	82.29 ^e^ ± 2.31	113.78 ^f^ ± 2.94	52.37 ^cd^ ± 1.03
Total polyphenols [g·100 g^−1^ d.w.]						2.08 ^b^ ± 3.16	18.86 ^j^ ± 2.56	16.32 ^h^ ± 1.89	3.15 ^c^ ± 1.55	9.15 ^f^ ± 3.03	1.56 ^a^ ± 2.81	6.00 ^e^ ± 2.44	10.70 ^g^ ± 2.11	4.08 ^cd^ ± 1.38

Data are expressed as mean values (n = 3) ± SD; SD—standard deviation. Mean values within rows with different letters are significantly different (*p* < 0.05).

## Data Availability

Not applicable.

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
