# Peer review of "Effects of Ozone Application on Microbiological Stability and Content of Sugars and Bioactive Compounds in the Fruit of the Saskatoon Berry (Amelanchier alnifolia Nutt.)"

_molecules, 2022, doi:10.3390/molecules27196446_

Round 1
Reviewer 1 Report
Review of the manuscript: Effects of ozone application on microbiological stability and content of sugars and bioactive compounds in the fruit of the
Saskatoon berry (Amelanchier alnifolia Nutt.)
In this study, present the effects of applied gaseous ozone at 10 ppm for 15 and 30 minutes on microbiological stability, sugar content and bioactive compounds for three cultivars and three clones of Saskatoon berry fruit.
1. I strongly suggest authors to introduce more keywords. The usefulness of keywords is to make the article both more and more easily searchable visible after its publication through commonly used search engines.
2. Introduction: The introduction is interesting, but in my opinion, it does not fully cover the topic. Below are some suggestions on how to expand this section. Moreover, out of 18 cited items, as many as 7 are older than 10 years. Can these items not be replaced with newer ones?
3. It must be pointed that only Folin-Ciocalteu method was used to identify the polyphenolic/phenolic content of the samples. It is a good way for comparing samples, but not to characterize them. More accurate methods could have been used.
believe that the work presented for review is of a high technical level, but it requires substantive amendments (please post new items).
I am asking for a deeper description, taking into account my suggestions above, with post new items.
Author Response
The authors are grateful for the contribution of the Reviewer.
- I strongly suggest authors to introduce more keywords. The usefulness of keywords is to make the article both more and more easily searchable visible after its publication through commonly used search engines.
Answer:
More keywords have been added to the article.
2. Introduction: The introduction is interesting, but in my opinion, it does not fully cover the topic. Below are some suggestions on how to expand this section. Moreover, out of 18 cited items, as many as 7 are older than 10 years. Can these items not be replaced with newer ones?
Answer:
The introduction was refined and 7 relatively old items of the cited literature were replaced with a relatively new one.
3. It must be pointed that only Folin-Ciocalteu method was used to identify the polyphenolic/phenolic content of the samples. It is a good way for comparing samples, but not to characterize them. More accurate methods could have been used.
Answer:
In the point of determination of bioactive compounds it is described in more detail that the total polyphenol content was determined by the Folin-Ciocalteu method.
Reviewer 2 Report
In this study, the author reporte'd the ozonation process had a positive effect on reducing the microbial load of the Saskatoon berry fruit. Besides the microbial tests, sugar and bioactive compounds content as well as antioxidant activity were also tested. Overall, this manuscript collected enough data to support the major conclusion and demonstrated high work-loading. However, before being accepted for publication, there is a major improvement that cannot be avoided. 1. All the data illustration methods need to be improved. Please draw a figure for comparison rather than list all of them as a table. It would be better to summarize major information as a figure and put the other as supplement material. 2. Language still need improvement. Please go through the grammar carefully before resubmitting after revision. 3. Please rewrite your abstract and conclusion. Be straightforward and clear.
Author Response
The authors are grateful for the contribution of the Reviewer.
- All the data illustration methods need to be improved. Please draw a figure for comparison rather than list all of them as a table. It would be better to summarize major information as a figure and put the other as supplement material.
Answer:
Tables 1, 2 and 5 contain too much information to show them in the form of figures (the graphs are illegible), but two figures have been added (Figures 3 and 4), which illustratively show how the content of ascorbic acid and total polyphenols changes depending on cultivar / clone and time of exposure to ozone.
2. Language still need improvement. Please go through the grammar carefully before resubmitting after revision.
Answer:
The article has been corrected by a native speaker.
3. Please rewrite your abstract and conclusion. Be straightforward and clear.
Answer:
The abstractand conclusions have been revised to present the results of our work more accurately and legibly.
Round 2
Reviewer 2 Report
All comments have been addressed.